# Contacts-based prediction of binding affinity in protein–protein complexes

**Anna Vangone, Alexandre MJJ Bonvin***

Computational Structural Biology Group, Bijvoet Center for Biomolecular Research, Faculty of Science—Chemistry, Utrecht University, Utrecht, Netherlands

**Abstract** Almost all critical functions in cells rely on specific protein–protein interactions. Understanding these is therefore crucial in the investigation of biological systems. Despite all past efforts, we still lack a thorough understanding of the energetics of association of proteins. Here, we introduce a new and simple approach to predict binding affinity based on functional and structural features of the biological system, namely the network of interfacial contacts. We assess its performance against a protein–protein binding affinity benchmark and show that both experimental methods used for affinity measurements and conformational changes have a strong impact on prediction accuracy. Using a subset of complexes with reliable experimental binding affinities and combining our contacts and contact-types-based model with recent observations on the role of the non-interacting surface in protein–protein interactions, we reach a high prediction accuracy for such a diverse dataset outperforming all other tested methods.

*For correspondence: a.m.j.j. bonvin@uu.nl

**Competing interests:** The authors declare that no competing interests exist.

## Introduction

Interactions between proteins play a central role in the processes happening in the cells, from DNA replication to protein degradation (*Jones and Thornton, 1996*; *Alberts, 1998*; *Nooren and Thornton, 2003*; *Perkins et al., 2010*). Perturbations in those interaction networks can lead to disease (*Stites, 1997*; *Sugiki et al., 2014*). Characterizing these protein–protein interactions (PPIs) is therefore crucial for a proper understanding of mechanisms in biological processes for disease research and for drug development, as most common targets of drugs are proteins (such as enzymes, ion channels, and receptors).

Exploring recognition processes at atomic level requires knowledge of the three-dimensional (3D) structure of the associated molecular complexes. It is however the binding affinity (BA) (i.e., the natural inclination of molecules to associate) that defines whether or not complex formation will occur. The BA is therefore the key for understanding and predicting recognition, association and dysfunction phenomena related to protein complexes. It has been shown that changes in BA caused by mutations or post-translational modification errors lead to various diseases (*Vidal et al., 2011*). Commonly, the affinity of an interaction is described through the equilibrium dissociation constant $K_d$, or, in thermodynamic terms, the Gibbs free energy $\Delta G$ ($\Delta G = RT \ln K_d$). Measuring $K_d$ values experimentally is a time-consuming and expensive process. Many computational methods aimed at predicting BA have been developed. Gaining the ability to predict BA is indeed an urgent need as it offers great opportunities not only to control interactions and develop innovative therapeutics (*Keskin et al., 2005*; *Aloy and Russell, 2006*; *Beltrao et al., 2007*; *Kiel et al., 2008*; *Dell'Orco, 2009*), but also for other applications such as protein engineering (*Kortemme et al., 2004*; *Sharabi et al., 2011*), computational mutagenesis (*Ben-Shimon and Eisenstein, 2010*), and docking (*Halperin et al., 2002*).

Different methods aimed at predicting BA have been proposed throughout the years, varying greatly in terms of accuracy and computational cost. Exact methods such as free energy

**eLife digest** Proteins help to copy DNA, transport nutrients and perform many other important roles in cells. To perform these tasks, proteins often interact with other proteins and work together. These interactions can be very complex because each protein has a three-dimensional shape that may change when it binds to other proteins. Also, two proteins may form several connections with each other.

It is possible to carry out experiments to calculate how likely it is that two proteins will physically interact with each other and how strong their connections will be. However, these measurements are time consuming and costly to do. Some researchers have developed computer models to help predict the interactions between proteins, but these models are often incorrect because they leave out some of the chemical or physical properties that influence the ability of proteins to interact.

With the aim of making a better model, Vangone and Bonvin examined 122 different combinations of proteins whose abilities to interact had previously been experimentally measured. Vangone and Bonvin found that the number of connections between each pair of proteins was a strong predictor of how tightly the proteins connect to each other. Particular features of the surface of the proteins—specifically, the region defined as the non-interacting surface—can also influence how strong the interaction is.

Vangone and Bonvin used this information to develop a new model that predicts how tightly proteins interact with each other based on the number of connections between the two proteins and the characteristics of the non-interacting surface. The model is simple, and Vangone and Bonvin show that it is more accurate than previous models. Defects in the interactions between proteins can lead to many diseases in humans, so this model may be useful for the development of new drugs to treat these conditions.

perturbation and thermodynamics integration can be very accurate, but due to their computational costs their application is extremely limited (mostly to low throughput studies and mainly for small drug binding or mutations). Methods based on empirical functions (empirical, force-field-based potentials, statistical potentials, and scoring functions used in docking) are much faster (*Jiang et al., 2002*; *Ma et al., 2002*; *Zhang et al., 2005*; *Audie and Scarlata, 2007*; *Su et al., 2009*; *Bai et al., 2011*; *Moal et al., 2011*; *Qin et al., 2011*; *Moal and Bates, 2012*; *Tian et al., 2012*; *Kastritis et al., 2014*; *Luo et al., 2014*). However, even if some have been very successful on small training sets (*Horton and Lewis, 1992*; *Audie and Scarlata, 2007*), most published models still fail to systematically predict BA (*Kastritis and Bonvin, 2010*) for large datasets or discriminate between binders and non-binders (*Sacquin-Mora et al., 2008*; *Fleishman et al., 2011*). The main weaknesses of these methods are that they usually neglect factors such as conformational changes upon binding, allosteric regulation, and solvent and co-factor effects, which may all contribute to the binding strength.

Binding between two proteins is mainly defined by their contact region, the interface, and it is indeed the network of contacts between surface residues that holds complexes together, defines their specificity and contributes to their interaction strength. The importance of such inter-residue contacts has already been established in docking. In the Critical Assessment of Prediction of Interactions (CAPRI) experiment (*Janin et al., 2003*), for instance, assessment of the accuracy of the docked models is based on a combination of positional root mean square deviation (RMSD) criteria and conservation of intermolecular contacts with respect to the native structure (*Lensink et al., 2007*). In the context of scoring, considering the conservation of contacts at the interface across docking models has been shown to improve the ranking of docked models (*Oliva et al., 2013*; *Chermak et al., 2014*). The atom contact frequency in a set of predictions, a similar concept, has also been included in the ZRANK docking pipeline (*Hwang et al., 2010*). Next to their use in scoring, contacts have been introduced as a way to cluster docking solutions based on the fraction of common inter-residue contacts among a set of decoys (*Rodrigues et al., 2012*).

However, in addition to properties of the interface, a recent work has also demonstrated an effect of the non-interacting surface (NIS) on BA (*Kastritis et al., 2014*), a finding that has been corroborated in a recent study by Marillet et al. (*Marillet et al., 2015*).

Here we show how the network of contacts at the interface of a protein–protein complex can help in describing the BA of the interaction. Based on the number contacts at the interface, we propose an innovative and very simple method to predict BA. To this end we used the protein–protein BA benchmark (*Kastritis et al., 2011*) consisting of 144 non-redundant protein–protein complexes with experimentally determined $K_d$ ($\Delta G$) and available 3D structures.

Our results show that interfacial contacts, which have so far been neglected in the rationalization and prediction of BA, can be considered the best structural property to describe binding strength. Based on this observation, we describe an extremely simple BA predictor that accounts for different types of contacts and shows the best performance reported so far (to our knowledge) on such a large and diversified dataset of complexes. Its performance is compared to other previously published predictor methods (*Moal et al., 2011*). Further, we analyze the impact of the experimental method used to characterize BA on the prediction performance.

## Results and discussion

Considering the critical role of the BA in the study of protein–protein complexes and the still elusive approaches to predict it (*Kastritis and Bonvin, 2010*; *Fleishman et al., 2011*), we demonstrate here that the network of inter-residue contacts (ICs) between two interacting proteins is a good descriptor for the BA. Using the structure-based BA benchmark of *Kastritis et al. (2011)*, we correlated ICs with experimentally determined BA data ($K_d$ or $\Delta G$) for protein complexes (bound forms). The diversity of experimental methods used to measure BA in this benchmark allows us to underline their limitations and reliability for use in BA prediction. From the original dataset of 144 complexes, we removed the cases with ambiguity in the exact $K_d$ values and all complexes with missing or unresolved residues (>2) at the interface (data reported in *Supplementary file 1*). For the remaining 122 complexes, we calculated the number of ICs and evaluated their correlation (expressed through the Pearson's correlation coefficient $R$) with the experimental $\Delta G$s. We describe the influence of various properties on the results, such as the distance cut-off defining a contact and the experimental method used for BA measurement.

### Correlation between ICs and BA

In a protein complex the interactions are usually of relatively short range. Recently, however, *Kastritis et al. (2014)* revealed the unexpected contribution of NIS residues to BA. We therefore systematically evaluated the effect of the distance on defining the inter-residues network by varying the cut-off distance between 3 Å and 20 Å (see 'Materials and methods'). For each distance cut-off the number of ICs was correlated with the experimental $\Delta G$s. The results, reported in *Figure 1*, show that the highest correlation is achieved at a cut-off of 4.0 Å, ($R_{ICs} = -0.50$, $\rho < 0.0001$). This correlation decreases slowly until 8.0 Å ($R_{ICs} = -0.41$, $\rho < 0.0001$) and drops at higher distances. We also evaluated the ranking power of the ICs, expressed through the Spearman's correlation coefficient $S$, which follows the same trend as $R$ with slightly higher absolute values (*Figure 1*).

### How experimental BA methods affect the correlations

Many different experimental methods can be used to determine the $\Delta G$ of a protein–protein complex. Each presents different characteristics so that the measured $\Delta G$ values depend on the method used, its sensitivity and the strength of the interactions that are being measured. For the set of 122 complexes used in this work, ten different experimental methods have been used to detect the $\Delta G$s: stopped-flow fluorimetry (8 cases), surface plasmon resonance (SPR) (40 cases), high performance liquid chromatography (HPLC)/UV absorption spectroscopy (14 cases), sedimentation (1 case), radioligand binding (competitive binding experiments) (2 cases), potentiometry (1 case), reduction assay (1 case), isothermal titration calorimetry (ITC) (19 cases), inhibition assay (17 cases), fluorescence spectrophotometric assays (19 cases).

We analyzed separately the BAs from the various experimental methods with enough data points (≥8) and compared the prediction performance with the full data set. As reported in *Table 1*, the correlations between ICs and experimental $\Delta G$s increased to $R_{ICs} = -0.70$ ($\rho = 0.03$), $R_{ICs} = -0.53$ ($\rho = 0.0003$), $R_{ICs} = -0.65$ ($\rho = 0.006$) and $R_{ICs} = -0.55$ ($\rho = 0.006$) in the case of $\Delta G$s determined by stopped-flow fluorimetry, SPR, spectroscopic methods and ITC, respectively. For the 17 cases measured by inhibition assays and the 19 by fluorescence spectrophotometry techniques the

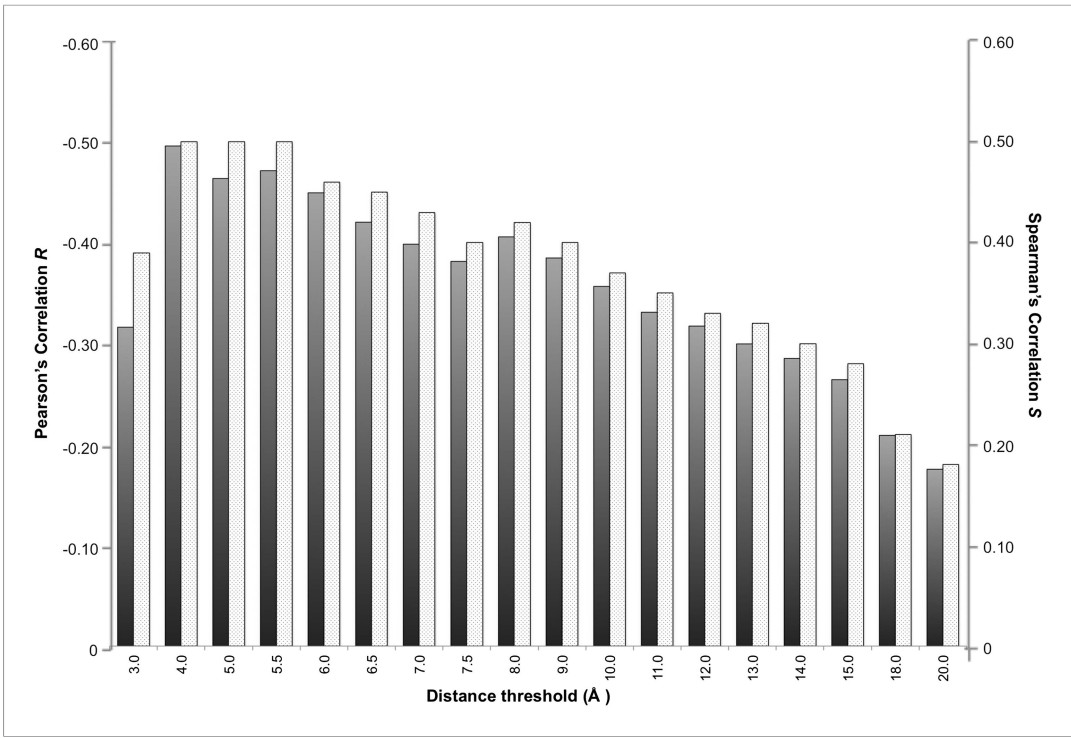

**Figure 1**. Correlation between number of inter-residue contacts and binding affinity ($\Delta$Gs) as a function of the distance cut-off used to calculate the contacts. Both the Pearson's *R* (dark grey bars) and the Spearman's *S* (light grey-patterned bars) correlation coefficient are reported.

correlations were meaningless ($R_{ICs}$ = 0.04 with $\rho$ = 0.9 and $R_{ICs}$ = 0.05 with $\rho$ = 0.8, respectively). These are indirect methods useful in calculating relative binding strengths (known as IC50s), but these have limitations when used to calculate absolute BA values (*Lazareno and Birdsall, 1993*; *Wilkinson, 2004*; *Masi et al., 2010*).

Removing the cases from inhibition assays and fluorescence spectrophotometry methods, and all others for which only a few data points were reported (potentiometry, radioligand, reduction assay, and sedimentation), we end up with a reliable dataset of 81 structures (a 'cleaned' dataset) showing an increased correlation of $R_{ICs}$ = −0.59 ($\rho$ < 0.0001) at the re-optimized distance threshold of 5.5 Å to define a contact (*Figure 2*). All further results will therefore refer to the 5.5 Å cut-off to define ICs.

**Table 1**. Pearson's correlations and p-values ($\rho$) between inter-residue contacts (ICs) and buried surface area (BSA) and experimental binding affinities ($\Delta$Gs) for the entire dataset and subsets corresponding to various experimental method

| Class | #Complexes | $R_{ICs}$ | | $R_{BSA}$ | |
|---|---|---|---|---|---|
| All | 122 | −0.50 | ($\rho$ < 0.0001) | −0.32 | ($\rho$ = 0.002) |
| Stopped-flow | 8 | −0.70 | ($\rho$ = 0.03) | −0.55 | ($\rho$ = 0.08) |
| SPR | 39 | −0.53 | ($\rho$ = 0.0003) | −0.44 | ($\rho$ = 0.002) |
| Spectroscopy | 14 | −0.65 | ($\rho$ = 0.006) | −0.27 | ($\rho$ = 0.2) |
| ITC | 20 | −0.55 | ($\rho$ = 0.006) | −0.64 | ($\rho$ = 0.001) |
| Inhibition assay | 17 | 0.05 | ($\rho$ = 0.4) | −0.08 | ($\rho$ = 0.4) |
| Fluorescence | 19 | 0.04 | ($\rho$ = 0.4) | 0.34 | ($\rho$ = 0.1) |

The ICs were calculated for a 4.0 Å cut-off.

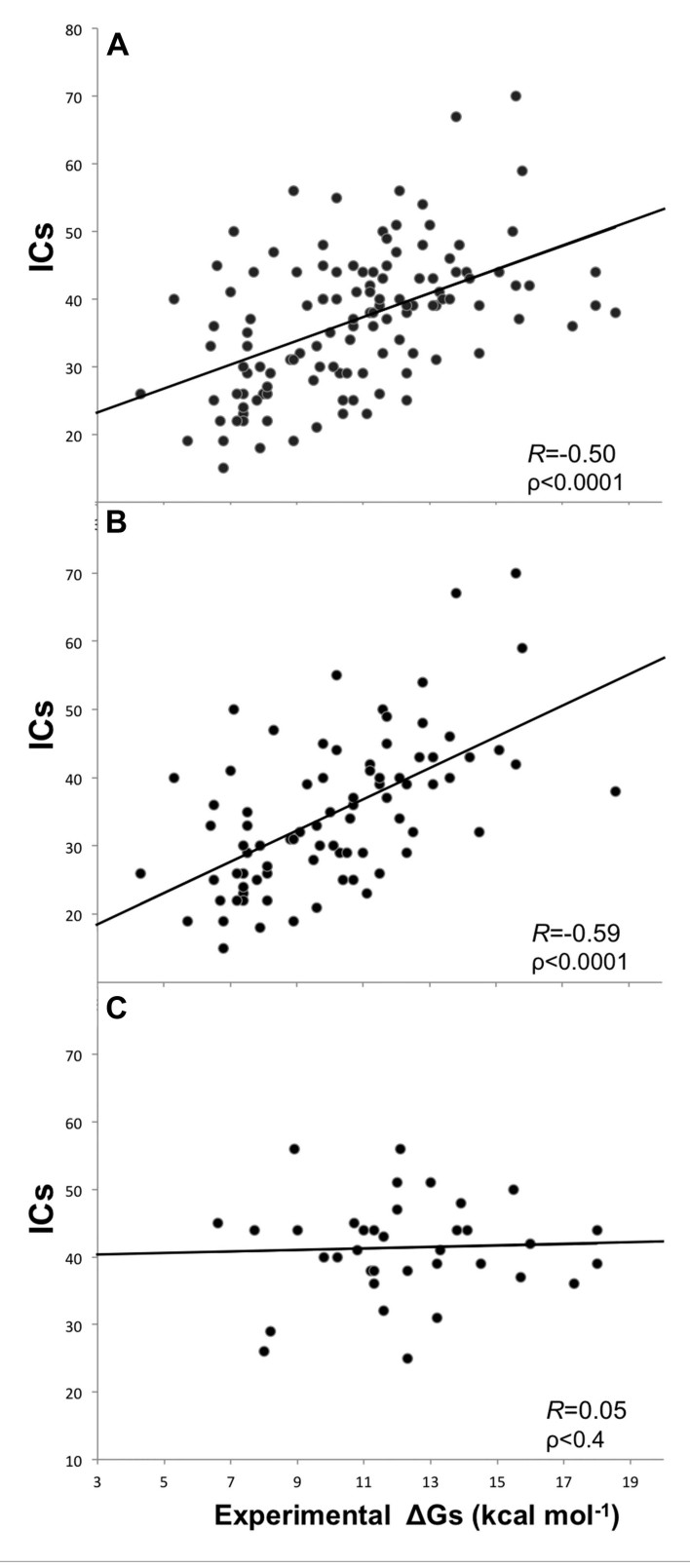

**Figure 2**. Plots of inter-residue contacts (ICs) vs experimentally determined binding affinities (ΔGs) of protein–protein complexes. (**A**) Full dataset (122 complexes), (**B**) reliable experimental methods only (stopped-flow, surface plasmon resonance, spectroscopy, isothermal titration calorimetry) (81 complexes), and (**C**) non-reliable experimental methods (inhibition assay and fluorescence) (36 complexes). The trend line and corresponding Pearson correlation coefficients and p-values (ρ) are reported in each plot; binding affinities are reported as absolute values.

## Structural properties contributing to BA

In order to assess which structural property might be the best descriptor for the binding strength, we calculated on the 'cleaned' dataset values for the widely used buried surface area (BSA), the NIS characteristics (recently shown to contribute in the BA) (*Kastritis et al., 2014*), and our ICs. We further classified these properties based on the amino acid type—polar/apolar—for BSA and NIS, and contact types—polar/polar, polar/charged, polar/apolar, charged/charged, charged/apolar, apolar/apolar—for the ICs. For the latter we also considered the hydrophobic/hydrophilic classification. For all these, we evaluated whether this finer classification (resulting, of course, in more parameters in our model) improves the correlations. It is clear from the results summarized in *Table 2* that the number of ICs is a better descriptor than the BSA, with $R_{ICs\_total} = -0.59$ ($\rho < 0.0001$) vs $R_{BSA\_total} = -0.46$ ($\rho < 0.0001$). When distinguishing between the amino acid properties, the factors that contribute the most to BA are the number of ICs between polar and apolar residues ($R = -0.56$, $\rho < 0.0001$) and between hydrophilic residues ($R = -0.53$, $\rho < 0.0001$). However, none of these individual classes shows better correlation than the total ICs and BSA. All the calculated data are provided in *Supplementary file 2*.

In order to assess the predictor power of the above-described structural properties, we built different predictor models (contacts-based, BSA-based, NIS-based and combinations of these), optimizing the following model:

$$\text{Model N}: \Delta G_{calc} = w_1 P_1 + w_2 P_2 + .... + Q, \tag{1}$$

where $P_N$ values are the properties used to train Model N, $w_N$ values are the weight and $Q$ is a shift value. To avoid the problem of over-fitting when many variables are used (>3), we applied the Akaike's Information Criterion (AIC) stepwise selection method implemented in R (*R Development Core Team, 2014*) in order to identify (and consider only) the significant variables among the training ones. Each derived model, with associated weights $w_N$ and performance, is reported in *Table 3*.

Models 1 and 2 were trained on ICs_total and BSA_total, respectively, with a better performance of the ICs-based model (as already reported in *Table 2*) (root mean square errors [RMSEs] of 2.25 and 2.46 kcal/mol, for ICs and BSA, respectively). Models 3 and 4 have been trained using the ICs classified by residue type (polarity in Model 3, hydrophobicity in Model 4). While single amino-acid-type ICs properties do not improve the correlations, their linear combination results in a significant improvement from $R = -0.59$ ($\rho < 0.0001$) for Model 1 to $R = -0.67$ ($\rho < 0.0001$) for Model 3 and R = −-0.60 for Model 4 ($\rho < 0.0001$). Model 5 has been trained on the polar/apolar classification of the Horton and Lewis BSA model, (*Horton and Lewis, 1992*), with a slightly improved performance compared with Model 2 based on the total BSA, but it is still worse than any of the contact-based models (i.e., Model 1, Model 3, and Model 4).

Among the models that are based on properties of the *interface* of the binding site, the one based on polarity-classification of ICs (i.e., Model 3) shows the best performance. We therefore added to it the *NIS* properties in order to obtain a full description of the structural properties of

**Table 2**. Pearson's correlations and p-values between experimental binding affinities and the inter-residue contacts (ICs), buried surface area (BSA) and non-interacting surface (NIS) (*Kastritis et al., 2014*) properties calculated on the 'cleaned' dataset

| Property | R | p-value |
|---|---|---|
| **ICs_total** | **−0.59** | **<0.0001** |
| ICs_charged/charged | −0.17 | =0.06 |
| ICs_charged/polar | −0.26 | =0.009 |
| ICs_charged/apolar | −0.45 | <0.0001 |
| ICs_polar/polar | −0.13 | =0.1 |
| ICs_polar/apolar | −0.56 | <0.0001 |
| ICs_apolar/apolar | −0.34 | =0.001 |
| ICs_hydrophilic/hydrophilic | −0.53 | <0.0001 |
| ICs_hydrophilic/hydrophobic | −0.34 | =0.001 |
| ICs_hydrophobic/hydrophilic | −0.05 | =0.3 |
| BSA_total | −0.46 | <0.0001 |
| BSA_polar | −0.36 | =0.0005 |
| BSA_apolar | −0.47 | <0.0001 |
| %NIS_polar | 0.07 | =0.06 |
| %NIS_apolar | −0.33 | =0.001 |
| %NIS_charged | 0.28 | =0.006 |

A fine classification of those properties based on the polar/apolar/charged and hydrophobic/hydrophilic nature of the amino acids is also reported. The property with the highest R value is highlighted in bold. The corresponding data are provided in **Supplementary file 2**.

**Table 3.** Optimization of binding affinity predictor models based on the regression model $\Delta G_{calc} = w_1 P_1 + w_2 P_2 + \ldots + Q$

| Properties ($P_N$) | Model 1 | Model 2 | Model 3 | Model 4 | Model 5 | Model 6 |
|---|---|---|---|---|---|---|
| ICs_total | 0.07782 | - | - | - | - | - |
| ICs_charged/charged | - | - | / | - | - | 0.09420 |
| ICs_charged/polar | - | - | / | - | - | / |
| ICs_charged/apolar | - | - | 0.11627 | - | - | 0.10038 |
| ICs_polar/polar | - | - | −0.12655 | - | - | −0.19522 |
| ICs_polar/apolar | - | - | 0.23595 | - | - | 0.22609 |
| ICs_apolar/apolar | - | - | / | - | - | / |
| ICs_hydrophil/hydrophil | - | - | - | 0.09055 | - | - |
| ICs_hydrophil/hydrophob | - | - | - | 0.05726 | - | - |
| ICs_hydrophob/hydrophil | - | - | - | 0.06037 | - | - |
| BSA_total | - | 0.00278 | - | - | - | - |
| BSA_polar | - | - | - | - | 0.00131 | - |
| BSA_apolar | - | - | - | - | 0.00400 | - |
| %NIS_polar | - | - | - | - | - | / |
| %NIS_apolar | - | - | - | - | - | −0.18786 |
| %NIS_charged | - | - | - | - | - | −0.13872 |
| Intercept (Q) | 4.78839 | 5.66032 | 5.13766 | 4.90452 | 5.44809 | 15.9433 |
| | | | | | | |
| R | −0.59 | −0.46 | −0.67 | −0.60 | −0.48 | −0.73 |
| p-value | <0.0001 | <0.0001 | <0.0001 | <0.0001 | <0.0001 | <0.0001 |
| RMSE (kcal mol⁻¹) | 2.25 | 2.46 | 2.08 | 2.22 | 2.45 | 1.89 |

The weights $w_N$ are reported for each properties $P_N$ used to train Model N. Properties that have not been used for training the Model or have been evaluated as not relevant from the Akaike's An Information Criterion (AIC) evaluation are reported as '-' and '/', respectively. At the bottom of the table, the correlation coefficient and prediction error (expressed as $R$ and RMSE) are reported.

the complex, resulting in Model 6. After AIC evaluation, we obtained the following linear equation:

$$\Delta G_{calc} = 0.09459\ ICs_{charged/charged} + 0.10007\ ICs_{charged\_apolar} - 0.19577\ ICs_{polar/polar}$$
$$+ 0.22671\ ICs_{polar/apolar} - 0.18681 \%NIS_{apola}r - 0.13810 \%NIS_{charged}$$
$$+ 15.9433.$$

$$(2)$$

Fourfold cross-validation results for this model (repeated 10 times) are reported in *Supplementary file 3*, showing consistency in terms of coefficient and prediction accuracy. A scatter plot of predicted vs experimental affinities is reported in *Figure 3*. The most relevant contributions to BA are the number of ICs made by charged and polar residues (ICs_charged/charged and ICs_polar/polar in *Equation 2*), while the apolar residues are only counted when interacting with charged and polar ones (ICs_charged/apolar and ICs_polar/apolar in *Equation 2*). This ICs/NIS-based model show the best performance of any model developed so far, with $R = -0.73$ and RMSE = 1.89 kcal mol⁻¹.

## The effect of conformational changes on BA-prediction accuracy

In many assemblies, the structure of free monomers differs from their structure in the oligomeric state (the 'bound' form) due to the association process. The affinity benchmark also reports the interface RMSD (i_rmsd) between the unbound and bound structures. This is a measure of how much

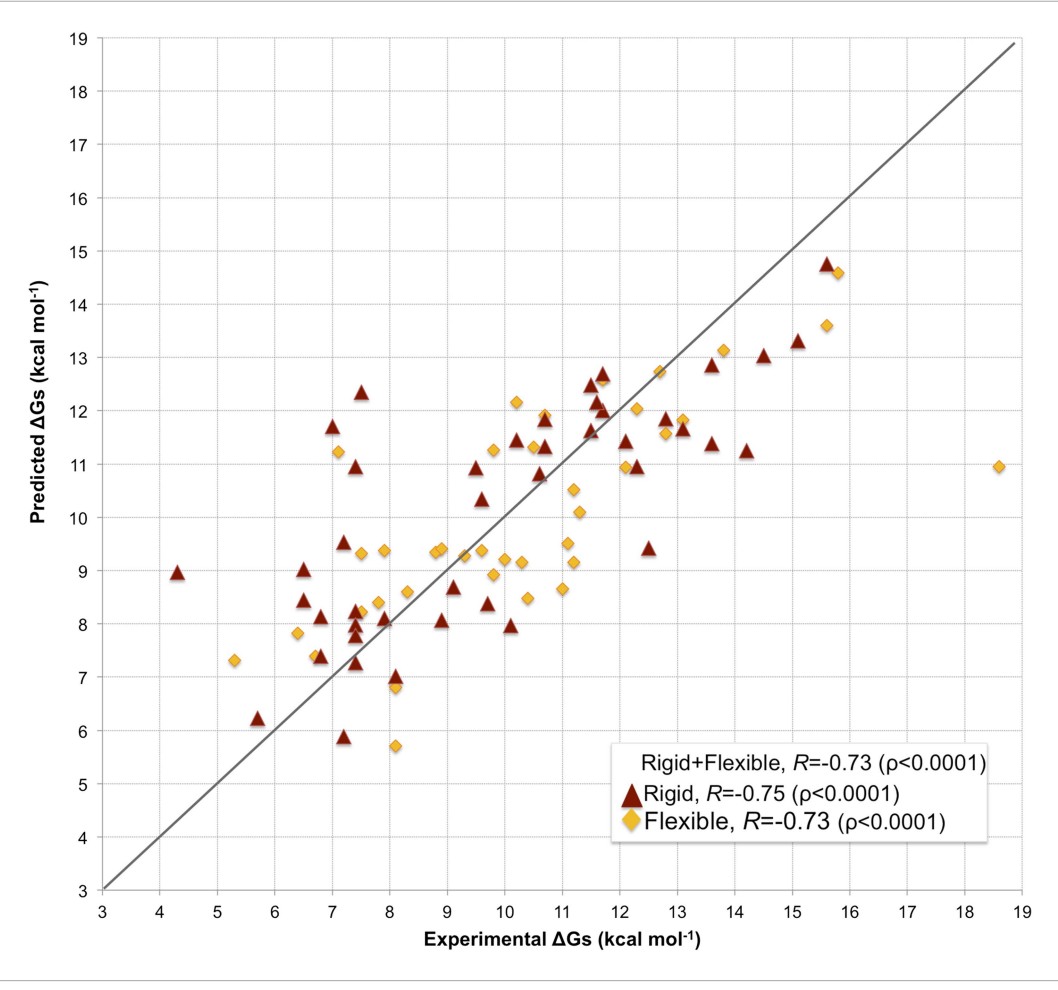

**Figure 3**. Scatter plot of predicted vs experimental binding affinities. The predictions were made according to the inter-residue contacts (ICs)/non-interacting surface (NIS)-based model (Model 6, *Equation 2*) for the cleaned dataset of 81 protein–protein complexes. The correlation for all 81 complexes yields an *R* of −0.73 ($\rho$ < 0.0001) with a RMSE of 1.89 kcal mol$^{-1}$. When only rigid cases (interface RMSD between superimposed free and bound components ≤1.0 Å, red triangles) are considered, the correlation increases to $R = -0.75$ ($\rho$ < 0.0001) with a RMSE of 1.88 kcal mol$^{-1}$, while for flexible cases (interface RMSD >1.0 Å; yellow rhombus) $R = -0.73$ ($\rho$ < 0.0001) with a RMSE of 1.88 kcal mol$^{-1}$. The x = y line is shown as reference; binding affinities are reported as absolute values.

conformational change takes place upon association. We investigated if our model would have a higher predictive power when classifying the complexes according to their amplitude of conformational change upon binding. Predictions made with our combined contacts and NIS model (Model 6, *Equation 2*) are much less sensitive to conformational changes than all previous models (*Figure 3* and *Figure 4*), with only minor differences in performance between rigid (i_rmsd ≤ 1.0 Å, $R = -0.75$ and RMSE = 1.88 kcal mol$^{-1}$) and flexible cases (i_rmsd > 1.0 Å, $R = -0.73$ and RMSE = 1.89 kcal mol$^{-1}$). This indicates that, in contrast to previous predictors, the number of interface residue contacts is a rather robust predictor that is less sensitive to conformational changes.

In order to perform a fair comparison with other previously published methods, we calculated their performance on the complexes that are in common between our clean dataset, the one reported by *Moal et al. (2011)*, and the pre-calculated data on the Computational Characterisation of Protein–Protein Interactions (CCHarPPI) web server, ending up in 79 protein–protein complexes (*Figure 4*). The considered models include the 'global surface model' of *Kastritis et al. (2014)*, the BSA-based model of *Horton and Lewis (1992)*, the top three best performing methods reported

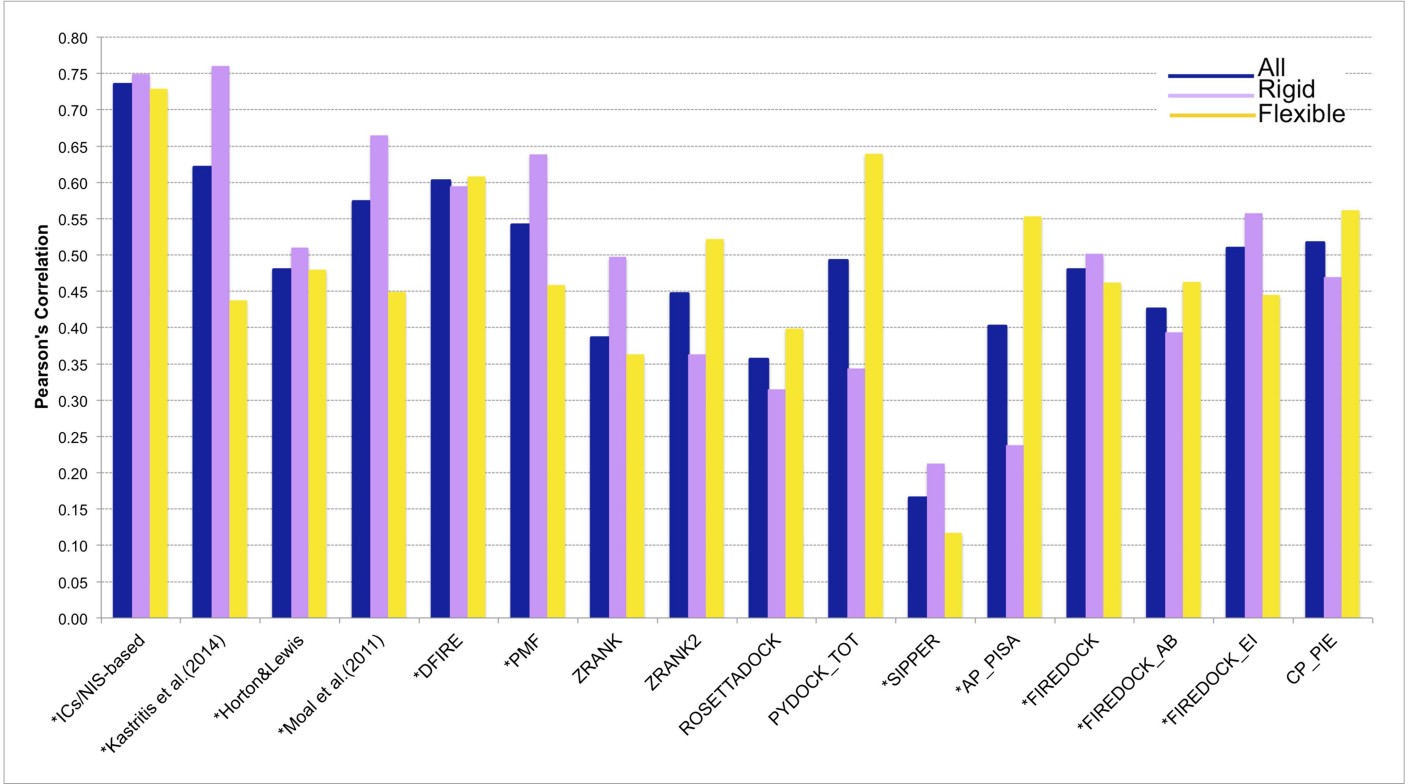

**Figure 4**. Comparison of the performance of our ICs/NIS-based model (Model 6, *Equation 2*) with other predictor models reported by *Moal et al. (2011)* and the CCHarPPI (*Moal et al., 2015a*, *b*) webserver. The performance is expressed as Pearson's Correlation coefficient between experimental and predicted binding affinities. Predictions were made on the common set of 79 complexes between our cleaned dataset, the data tested by *Moal et al. (2011)* and the CCHarPPI (*Moal et al., 2015a*, *b*) pre-calculated data. Correlations for the entire set and the rigid (43) and flexible (36) complexes are reported as absolute values for easier comparison (methods marked with asterisk showed original negative correlations).

by *Moal et al. (2011)* (their consensus model, DFIRE [*Liu et al., 2004*] and PMF[*Su et al., 2009*]) and the composite scoring functions reported by the CCHarPPI webserver [*Moal et al., 2015a*, *b*], such as ZRANK [*Pierce and Weng, 2007*], ZRANK2 [*Pierce and Weng, 2008*], RosettaDock [*Chaudhury et al., 2010*], PyDock [*Cheng et al., 2007*], FireDock [*Andrusier et al., 2007*], PISA [*Viswanath et al., 2013*], PIE [*Ravikant and Elber, 2010*], and SIPPER [*Pons et al., 2011*]. As shown in *Figure 4*, our ICs/NIS-based model (Model 6) outperforms all other methods tested. It is also less sensitive to conformational changes. All associated data are provided in *Supplementary file 4*. In addition to the composite scoring function of CCHarPPI, none of the other 99 intermolecular parameters reported by CCHarPPI outperformed our model, even if some show correlations above −0.50.

## Insights on the difference between ICs and BSA

The ICs introduced in this work to describe BA seem to be a 'higher level definition' structural parameter than the BSA since they express not only the contact surface but also the pairwise non-bonded interactions that the two proteins make, which is related to the packing of the interface. Indeed, a weak complex is expected to be less tightly packed than a strong one, a difference that should be better reflected in the ICs than in the BSA. In particular, the origin of this difference might reside in the fact that the contribution of each residue to the BSA will greatly depend on the solvent-accessible surface area of the residue in the free form, whereas this does not affect ICs. To illustrate this point we checked the main differences between ICs and BSA for the complex between Fab D3H44 and Tissue factor (PDB code: 1JPS [*Faelber et al., 2001*]; $\Delta G_{exp}$ = −13.6 kcal mol$^{-1}$). This complex has a BSA of 1852 Å$^2$ and 83 ICs at 5.5 Å, resulting in a ICs-based affinity prediction of

−12.8 kcal mol$^{-1}$ and a BSA-based one of −10.3 kcal mol$^{-1}$. The relative contribution of each interfacial Fab residue to the total BSA and number of ICs is shown in *Figure 5A—figure supplement 1A*: The contributions of the various residues to the ICs are more equally distributed than for the BSA. The latter shows high contributions for some residues, which is closely related to their solvent-accessible surface area (ASA) in the free form (defined here as the conformation extracted from the complex, that is, without any conformational changes—see 'Materials and methods') (see *Figure 5B—figure supplement 1*) and the surface representation in *Figure 5*. Indeed, because BSA$_{Fab}$ = ASA$_{Fab\_free}$ −ASA$_{Fab\_complex}$, when a residue is at the core of the binding interface (in other words totally shielded by Tissue factor) the ASA$_{Fab\_complex}$ will be close to 0, resulting in BSA$_{Fab}$ ~ASA$_{Fab\_free}$. In contrast, residues already almost fully buried in the free form will not contribute to the BSA, whereas they might be able to form contacts in our ICs model.

To report an example of this, the three residues GLU_H31, TYR_H32, and TYR_H33 located at the core of binding site (see *Figure 5*) contribute equally to the ICs calculation (6%), while they account for 10.5%, 5.6%, and 8.1% of the BSA$_{Fab}$, respectively. On the other hand, residues such as ILE_H29, TYR_H53 and ASP_H99, which are already highly buried in the free form and therefore contribute less than 0.1% to the BSA$_{Fab}$, are still making contacts, contributing between 1.2% and 2.4% in the calculation of ICs. In general, it seems that our ICs-based model is accounting more evenly than the BSA model for the contributions of highly solvent-accessible and buried residues, which leads to a higher prediction accuracy.

## Concluding remarks

Our new ICs/NIS-based BA model predicts BAs with an unprecedented accuracy ($R = -0.73$, $\rho < 0.0001$; RMSE = 1.89 kcal mol$^{-1}$), on a large, various and reliable dataset of 81 complexes. It achieves this by making use of only two structural features: the interfacial residue–residue contacts and the contribution of the NIS. Unsurprisingly, the higher the number of interfacial contacts, the stronger the binding strength, which is consistent both with the previously reported evidence that interfaces tend to be larger and more tightly packed with increasing interaction strength (*Nooren and Thornton, 2003*) and with the simple BSA models introduced by *Chothia and Janin (1975)* and *Horton and Lewis (1992)*. BSA and the number of contacts at the interface are of course somewhat related, but the number of interface contacts shows much better correlations with binding strength than the BSA.

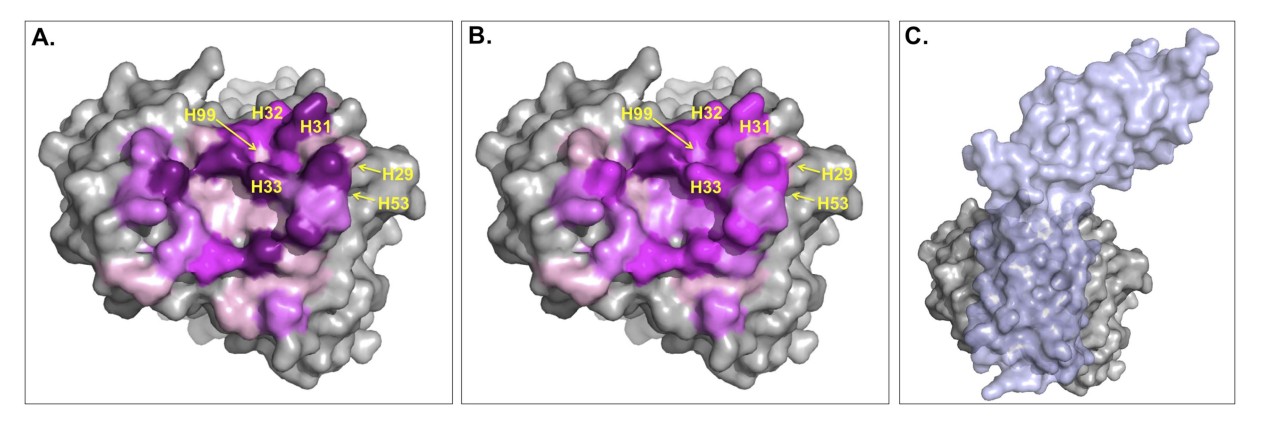

**Figure 5**. Surface representation of Fab D3H44; residues at the interface are colored according to their contribution (in percentage) to **(A)** the buried surface area (BSA) of Fab upon complex formation and **(B)** the total number of inter-residue contacts (ICs) made. Increasing graduation of pink is used for the ranges 0–2%, 2–4%, 4–6%, and above 6% of BSA/ICs contribution. **(C)** Surface representation of Fab D3H44 (gray) in complex with Tissue factor (light blue), PDB code: 1JPS (*Faelber et al., 2001*). Fab D3H44 is represented in all panels with the same orientation. Values of residues BSA/ICs contribution are reported in *Supplementary file 5*. The following figure supplement is available for *Figure 5*.

The following figure supplement is available for figure 5:

**Figure supplement 1**. Comparison between BSA and ICs relative contribution.

In summary, our study demonstrated that interface contacts, decomposed into their polar/apolar/ charged characteristics, and combined with contributions of the NIS based on the recent work of *Kastritis et al. (2014)* (in particular the percentage of apolar and charged surface), lead to the best BA predictor for protein–protein complexes reported to date. Importantly, these are less sensitive to conformational changes occurring upon binding, which are one of the challenging aspects to deal with for both structure and affinity prediction.

## Materials and methods

### Dataset

In order to evaluate the relationship between the contacts at the interface and the experimental BA in protein–protein complexes, we used the bound structures from the structure-based protein–protein BA benchmark of *Kastritis et al. (2011)*. It contains 144 non-redundant protein–protein complexes with known 3D structures (of both the unbound and bound components) and associated experimental ΔG values.

From this dataset we removed: (i) three cases (PDB codes: 1NSN, 1UUG, and 1IQD) for which the ΔG has not a unique value (reported as > −14 kcal/mol, > −18 kcal/mol, and > −15 kcal/mol, respectively), and (ii) all the complexes that show gaps or unresolved fragments at the binding interface (considering a gap to be a missing segment longer than two residues). 19 cases were discarded in total (for details see *Supplementary file 1*). This resulted in a dataset of 122 complexes, covering diverse types of biological functions including antibody/antigen (A or AB with bound antibody, 10 cases), enzyme/inhibitors (EI, 34 cases), enzyme/substrate (ES, 9 cases), enzyme/ regulatory subunit (ER, 7 cases), G-protein containing (OG, 15 cases), membrane receptors (OR, 7 cases), miscellaneous (OX, 26 cases), and non-cognate complexes (NC, 9 cases). The dataset includes both weak and strong complexes in terms of interaction strength, with ΔG values ranging between −4.3 and −18.6 kcal/mol.

The published benchmark also reports for each entry the interface C-alpha RMSD (i_rmsd) between unbound and bound form, which provides an estimate of the amplitude of the conformational changes that take place upon binding. Our clean dataset has i_rmsd ranging between 0.17 Å and 4.90 Å. The interacting area, expressed in terms of BSA upon complex formation, ranges from 808 Å$^2$ to 3370 Å$^2$.

### Contacts, BSA and NIS calculation

We calculated the number of interface residue pair-wise contacts (ICs) for each complex using the COCOMAPS web tool (*Vangone et al., 2011*). Two residues are considered in contact if a pair of (any) atoms belonging to two residues is closer than a defined cut-off distance. To systematically evaluate the impact of the cut-off distance on the correlation between ICs and BA, we varied the cut-off between 3 Å and 20 Å in steps of 0.5 Å in the range 5–8 Å and 1.0 Å from 8 Å and above.

The BSA upon complex formation was calculated using NACCESS (*Hubbard and Thornton, 1993*) as:

$$BSA = (ASA_{protein1} + ASA_{protein2}) - ASA_{complex}, \qquad (3)$$

where $ASA_{protein1}$ and $ASA_{protein2}$ are the solvent-accessible surface areas calculated from the free components (i.e., the separated bound conformation of the proteins—note that this is different from the unbound form of the protein) using a 1.4 Å radius sphere.

The NIS properties (i.e., percentage of polar, apolar and charged residues on the NIS) were calculated as described in *Kastritis et al. (2014)*.

Residues were classified based on their physico-chemical properties as follow:

- polar: C, H, N, Q, S, T, Y, W
- apolar: A, F, G, I, V, M, P
- charged: E, D, K, R.

The hydrophobic nature of the residues was defined according to the Kyte–Doolittle hydrophobicity index (*Kyte and Doolittle, 1982*).

The scripts for calculation of ICs (polar/apolar/charged divided) and NIS are available at: http:// bonvinlab.org/software. A description of how to predict binding affinity with our approach is described in details in Bio-Protocol (*Vangone and Bonvin, 2017*).

## Correlations, prediction power and cross-validation

To assess the linear dependence between two variables (i.e., the experimental BA and the structural properties tested, such as ICs), the Pearson product-momentum correlation coefficients ($R$) were calculated. The ranking power of ICs with BA was also calculated as reported by the Spearman's correlation parameter $S$.

We trained different models (*Equation 1*) using linear regression in R (*R Development Core Team, 2014*); to avoid problem of over-fitting when many variables were used (>3) we applied the AIC stepwise selection approach (backward and forward) in order to identify the significant terms and calculate weights only for them.

Cross-validation on the final model was performed by partitioning the dataset into four complementary subsets, training on the 75% of the data (training set) and validating on the other 25% (prediction set). Such fourfold cross-validation was repeated 10 times.

## Comparison with other methods

We compared our method with other BA predictors, potentials and composite scoring functions. Their performance is reported as correlation ($R$) between the predicted BA (or potential) and the experimental BAs. The comparison was made for 79 protein–protein complexes that are in common between our cleaned dataset of 81 structures and the 137 complexes reported by *Moal et al. (2011)*.

Predicted BAs for the 'global surface model' developed by *Kastritis et al. (2014)* have been calculated through the program provided by the authors; the Horton & Lewis BSA-based model (1992) (*Horton and Lewis, 1992*) was re-trained on our clean dataset (reported as Model 5); data for the consensus model reported by *Moal et al. (2011)*, DFIRE (*Liu et al., 2004*), and PMF (*Su et al., 2009*) are provided in (*Moal et al., 2011*); data of the composite scoring ZRANK (*Pierce and Weng, 2007*), ZRANK2 (*Pierce and Weng, 2008*), RosettaDock (*Chaudhury et al., 2010*), PyDock (*Cheng et al., 2007*), FireDock (the total energy, the antibody–antigen energy and the enzyme-inhibitor energy) (*Andrusier et al., 2007*), PISA (*Viswanath et al., 2013*), PIE (*Ravikant and Elber, 2010*), and SIPPER (*Pons et al., 2011*) were obtained as pre-calculated data from the CCHarPPI webserver (*Moal et al., 2015a, b*). Apart from the composite scoring function, CCHarPPI reports 99 additional intermolecular parameters, such as potential functions, energy functions, and various descriptors. The performance of each of them has been compared with our model.

## Acknowledgements

The authors thank Dr Gert Folkers, Dr Panagiotis Kastritis and Prof. Jeff Grinstead for valuable discussion, and Dr Li Xue for her highly valuable contribution to the statistical analysis.

## Additional information

### Funding

| Funder | Grant reference | Author |
|---|---|---|
| Seventh Framework Programme | WeNMR e-Infrastrure project, grant 261572 | Anna Vangone, Alexandre MJJ Bonvin |

The funder had no role in study design, data collection and interpretation, or the decision to submit the work for publication.

### Author contributions

AV, Conception and design, Acquisition of data, Analysis and interpretation of data, Drafting or revising the article; AMJJB, Conception and design, Analysis and interpretation of data, Drafting or revising the article

### Author ORCIDs

Alexandre MJJ Bonvin, http://orcid.org/0000-0001-7369-1322

## Additional files

### Supplementary files

• Supplementary file 1. List of entries removed from the original binding affinity benchmark (*Alberts, 1998*) because of gaps at interface.

• Supplementary file 2. ICs, NIS and BSA values (and their sub-classification) calculated on the cleaned dataset. Correlations of these to the binding affinity are reported in *Table 2*.

• Supplementary file 3. Table summarizing the weights ($w_N$) and performance (expressed as Pearson's coefficient $R$ and RMSE) of the fourfold cross-validation, repeated 10 times, of the following binding affinity regression model:

$$\Delta G_{calc} = w_1\, ICs_{charged/charged} + w_2\, ICs_{charged\_apolar} - w_3\, ICs_{polar/polar}$$
$$+ w_4\, ICs_{polar/apolar} + w_5\, \%NIS_{apolar} + w_6\, \%NIS_{charged} + Q.$$

Each coefficient has been reported as average on the four models trained on the respective folds.

• Supplementary file 4. Predicted binding affinities based on ICs (this work) and other methods (see 'Materials and methods') for the set of 79 common complexes between the cleaned dataset and the data tested reported by *Moal et al. (2011)*. Performance results are summarized in *Figure 4*.

• Supplementary file 5. List of Fab D3H44 antibody residues in the binding interface of the complex with the Tissue factor (PDB code of the complex: 1JPS). For each residue N, its relative contribution (expressed as a percentage) to the total number of inter-residue contacts made and to the total buried surface area of the Fab is reported. $IC_N$ and $BSA_N$ are the interface contacts and the buried surface area of residue N, respectively; $IC_{total}$ is 83; $BSA_{Fab\_total}$ is evaluated as half of the BSA for the complex corresponding to 926 Å$^2$.

### Major datasets

The following previously published datasets were used:

| Author(s) | Year | Dataset title | Dataset ID and/or URL | Database, license, and accessibility information |
|---|---|---|---|---|
| Kastritis PL, Moal IH, Hwang H, Weng Z, Bates PA, Bonvin AMJJ, Janin J | 2011 | A structure-based benchmark for protein–protein binding affinity | http://onlinelibrary.wiley.com/doi/10.1002/pro.580/suppinfo | Publicly available at the Wiley Online Library. |
| Moal IH, Jiménez-García B, Fernández-Recio J | 2015 | CCharPPI: Computational Characterisation of Protein–Protein Interactions | http://life.bsc.es/pid/ccharppi/info/affinity_benchmark | Publicly available at the Computational Characterisation of Protein–Protein Interactions. |

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
