## [Decision Letter]

Thank you for sending your work entitled “Contacts-based prediction of binding affinity in protein-protein complexes” for consideration at *eLife*. Your article has been favorably evaluated by Michael Marletta (Senior editor), a Reviewing editor, and two reviewers.

Both reviewers found the manuscript interesting and we would like to invite the authors to submit a revised manuscript that addresses the reviewers' comments.

*Reviewer #1*:

The authors wish to predict binding affinity (BA), clearly a problem whose importance is hard to overstate. This is a mature field, and has an excellent tradition of methodological contributions with discussion of physicochemical significance. The authors find that ICs are more strongly correlated to BA than buried surface area (BSA) is. No explanation is provided for this, which I think should be corrected. I also think since prior workers have separated BSA into polar and nonpolar contributions, counted hydrogen bonds, etc., the authors should do the same for ICs, as this may improve the results.

It is interesting that IC appears to be a better predictor of BA compared to BSA, even though IC seems a cruder measure. Please try to find the underlying reason for this. I suspect I know the reason.

The fact that the 4Å cutoff is optimal is significant, in my opinion. The authors should try to explain.

It is not clear that the minimum correlation is at 20Å, since this is the largest interface distance tested.

Mean absolute error is not a very standard error measure. Please use a measure that is more common in the literature.

The fact that only three fitting parameters were used effectively eliminates the possibility of overfitting. Still, I think the success of the method raises important questions. Do all contacts contribute equally to BA? What if contacts were to be grouped into e.g. salt bridges, hydrogen bonds, hydrophobic, etc.? I don't think it would be hard to count these. Horton and Lewis did something like this, separating polar from nonpolar buried surface area. I don't see that they report the error, but their Table 2 suggests it is comparable to that in the present work. Chothia and Janin also indicate the importance of separating polar and nonpolar contacts. The current results are not compared quantitatively to methods from the literature. Doing this would make it clear to what extent the new method is better than the existing ones.

In general I think this paper is promising. I don't doubt that it will eventually be published, either here or elsewhere. However the questions it raises ought to be answered. Otherwise this paper will merely report a phenomenon, without adding understanding. Also the comparison to past methods should be improved.

*Reviewer #2*:

It is an interesting work dealing with the prediction of energies of protein - protein interactions. Some aspects of the work should be improved.

The authors do not report the performance of other methods when applied to the same dataset. I see only the comparison of the performance of inter-residue contacts to buried surface area. The report of an explicit comparison of results obtained on the same dataset by other approaches, well quoted into the Introduction, should improve the quality of the article.

In the paragraph “The effect of conformation changes on BA prediction's accuracy” the authors refers again to other models to indicate their method as preferred for being less sensitive to conformational changes, without reporting a comparison of the results obtained by that models for the two subsets of rigid and flexible complexes.

The training and prediction sets give quite different results. This should be discussed and explained. Maybe, the random division in the two sets could be repeated a high number of times, and a mean result could be considered.

---

## [Author Response]

Reviewer #1:

*The authors wish to predict binding affinity (BA), clearly a problem whose importance is hard to overstate. This is a mature field, and has an excellent tradition of methodological contributions with discussion of physicochemical significance. The authors find that ICs are more strongly correlated to BA than buried surface area (BSA) is. No explanation is provided for this, which I think should be corrected. I also think since prior workers have separated BSA into polar and nonpolar contributions, counted hydrogen bonds, etc., the authors should do the same for ICs, as this may improve the results*.

*It is interesting that IC appears to be a better predictor of BA compared to BSA, even though IC seems a cruder measure. Please try to find the underlying reason for this. I suspect I know the reason*.

To shed light about the better performance of ICs vs BSA, we analyzed in detail some cases comparing differences between interface residues in terms of their their BSA and ICs. In the revisited manuscript we report the results of this analysis using as test case the complex between the Fab D3H44 and the Tissue factor, focusing the attention on Fab interface (PDB code; 1JPS); the new data are in paragraph “Insights on the difference between ICs and BSA”, including details in Figure 5, Figure 5—figure supplement 1, and [Supplementary-material SD5-data].

Considering one of the two interacting protein (but the same concepts applies to the other as well), both ICs and BSA parameters detect roughly the same residues at the interface, but when counting the contribution of each residue to the total BSA and ICs the results can greatly vary. To highlight this, we calculate the relative contribution (percentage) of each residue to the total BSA or number of ICs.

We found the following aspects:

The contribution of a residue to the ICs mostly depends on its position with respect to the binding partner rather than its solvent accessibility: every core residues (even those rather buried already in the free form), i.e. the ones positioned at the center of the binding site equally facing the interacting partner, have about the same contribution to the total ICs value (around 6% in the reported example case).

The contribution of each residue to the BSA highly depends on its ASA in the free form instead of its relative position with respect to the binding partner. Indeed, as BSA _protein_ = ASA_protein_free_ – ASA_protein_in_complex_, the core residues will be almost totally shielded by the interacting protein (means ASA_protein _ in_complex_ ∼ 0), resulting in BSA_protein_ ∼ ASA_protein_free_. This implies that residues already quite buried in the free form will have small contribution while they still might make several contacts. This results in an unequal contribution of residues in the interface (see new Figure 5). As an illustration, in the example case reported in the revisited paragraph, ASP_H52, which has a ASA of ∼ 20 Å^2^ in the free form, contributes poorly to the BSA while still forming 4 ICs. Further examples are reported in the text (subsection “Insights on the difference between ICs and BSA”). This dependency of the BSA on the ASA of the free form brings an unequal evaluation of core residues and an underestimation of partially buried residues located anyhow in an optimal position to contact the partner. The ICs seem less biased toward the size of the residue itself and more influenced by its orientation with respect to the binding partner.

*The fact that the 4Å cutoff is optimal is significant, in my opinion. The authors should try to explain*.

The 4Å cutoff is indeed close to the well accepted value of 3.9Å used to define non -bonded contacts for example in the ligplot/dimplot software. This optimal value was obtained considering the complete initial dataset; however, after removing the data collected with less reliable experimental methods we re- optimize such threshold up to 5.5 Å, at which we got the highest Pearson’s correlation coefficient (*R*) of -0.59. We then kept this 5.5Å cutoff for the further analysis on the reliable dataset.

This 5.5Å threshold allows us to include different non-bonded interactions, including salt-bridged (often counted up to 5Å cutoff).

*It is not clear that the minimum correlation is at 20Å, since this is the largest interface distance tested*.

Considering it is a new approach to correlate contacts with the binding affinity, we tested the impact of distance threshold on the correlation, checking correlations in the range between 3.0 Å and 20.0 Å. We observed the maximum correlation at 5.5 Å followed by a gradually drop off (reaching the low significant value of *R* = -0.18); therefore we would not expect an increasing of it after 20.0 Å. Also increasing the cutoff further would make the ICs dependent on the protein sizes as more contacts will be detected for larger proteins, even at similar interface sizes.

*Mean absolute error is not a very standard error measure. Please use a measure that is more common in the literature*.

We changed it reporting the more commonly used Root Mean Square Error (RMSE) (for instance as reported in Moal et al. Bioinformatics 2011).

*The fact that only three fitting parameters were used effectively eliminates the possibility of overfitting. Still, I think the success of the method raises important questions. Do all contacts contribute equally to BA? What if contacts were to be grouped into e.g. salt bridges, hydrogen bonds, hydrophobic, etc.? I don't think it would be hard to count these. Horton and Lewis did something like this, separating polar from nonpolar buried surface area. I don't see that they report the error, but their*
Table 2
*suggests it is comparable to that in the present work. Chothia and Janin also indicate the importance of separating polar and nonpolar contacts*.

This was an excellent suggestion! Following the reviewer suggestion, we checked if all residues contributed equally to the ICs parameter classifying them (and the resulting contact classes) according to their polar/apolar/charged character and to their hydrophobic/hydrophilic one. We trained multiple models using those different classifications. The results have been reported in the new paragraph “Structural properties contributing to binding affinity”*,* with updated Figure 3 and two new supplementary tables (Tables 2 and 3, [Supplementary-material SD2-data] and [Supplementary-material SD3-data]).

As foreseen by the reviewer, the correlations improve in all the classification considered. In particular, from a correlation of *R*= -0.59 when considering all the ICs, we obtained a correlation of -0.67 and -0.60 for the polarity- and hydrophobicity- based classification, respectively. Improvements have been observed also for the BSA, which *R* went from -0.46 to -0.48 when polarity classification has been considered.

Based on these new findings, we then trained the combined ICs/NIS (Non Interacting Surface, in Kastritis et al. JMB 2014) models using polarity-based ICs classification. From *R*= -0.67 obtained in the previous version of the work, we reached a correlation of *R*= -0.73 in this revisited version!

Finally, due to the fact that some of the tested models were trained on more than 3 parameters, we applied the AIC method to detect only the significant variables bypassing the risk of overfitting (details added in the Materials and methods section, paragraph “Correlations, prediction power and cross-validation”).

*The current results are not compared quantitatively to methods from the literature. Doing this would make it clear to what extent the new method is better than the existing ones*.

We added this part (subsection “The effect of conformational changes on binding affinity prediction accuracy” and Figure 4), providing comparisons with a large number of other methods (110 in total). From the plot it is clear that our method outperforms all reported ones and is also the most robust in case of conformational changes.

*In general I think this paper is promising. I don't doubt that it will eventually be published, either here or elsewhere. However the questions it raises ought to be answered. Otherwise this paper will merely report a phenomenon, without adding understanding. Also the comparison to past methods should be improved*.

We hope the new insights we added and the clear comparison with the past methods made clearer the new aspects (and performance) presented in the work.

Reviewer #2:

*It is an interesting work dealing with the prediction of energies of protein - protein interactions. Some aspects of the work should be improved*.

*The authors do not report the performance of other methods when applied to the same dataset. I see only the comparison of the performance of inter-residue contacts to buried surface area. The report of an explicit comparison of results obtained on the same dataset by other approaches, well quoted into the Introduction, should improve the quality of the article*.

We thank the reviewer for his/her comments that helped us improve the quality of our work. Comparison with existing methods have been added (as also asked by reviewer 1) and is reported in subsection “The effect of conformational changes on binding affinity prediction accuracy” and in Figure 4.

*In the paragraph “The effect of conformation changes on BA prediction's accuracy” the authors refers again to other models to indicate their method as preferred for being less sensitive to conformational changes, without reporting a comparison of the results obtained by that models for the two subsets of rigid and flexible complexes*.

The impact of conformational changes upon binding on the predicting power of our model has been reported in the paragraph “The effect of conformational changes on binding affinity prediction accuracy”, and in Figure 3, clearly showing the high stability of the performance even in cases of high conformational changes (defined “flexible”): *R* on total dataset = -0.73, *R* on rigid cases = - 0.75, *R* on flexible cases = -0.73. Results on the all, rigid and flexible cases for our and other tested methods are also reported in Figure 4, in which our method shows the highest stability (little difference in prediction performance).

*The training and prediction sets give quite different results. This should be discussed and explained. Maybe, the random division in the two sets could be repeated a high number of times, and a mean result could be considered*.

Considering the improved results we obtained after addressing Reviewer 1’s comments, we updated our model removing that part.